# PAST: The Pathway Association Studies Tool to Infer Biological Meaning from GWAS Datasets

**DOI:** 10.3390/plants9010058

**Published:** 2020-01-02

**Authors:** Adam Thrash, Juliet D. Tang, Mason DeOrnellis, Daniel G. Peterson, Marilyn L. Warburton

**Affiliations:** 1Institute for Genomics, Biocomputing & Biotechnology, Mississippi State University, Mississippi State, MS 39762, USA; peterson@igbb.msstate.edu; 2USDA-FS Forest Products Laboratory, Starkville, MS 39759, USA; juliet.d.tang@usda.gov; 3Humanities and Fine Arts Division, East Mississippi Community College, Mayhew, MS 39752, USA; mason.deornellis@gmail.com; 4Department of Computer Science and Engineering, Mississippi State University, Mississippi State, MS 39762, USA; 5USDA-ARS Corn Host Plant Resistance Research Unit, Mississippi State, MS 39762, USA

**Keywords:** metabolic pathway analysis, genome-wide association study (GWAS), maize (*Zea mays* L.)

## Abstract

In recent years, a bioinformatics method for interpreting genome-wide association study (GWAS) data using metabolic pathway analysis has been developed and successfully used to find significant pathways and mechanisms explaining phenotypic traits of interest in plants. However, the many scripts implementing this method were not straightforward to use, had to be customized for each project, required user supervision, and took more than 24 h to process data. PAST (Pathway Association Study Tool), a new implementation of this method, has been developed to address these concerns. PAST has been implemented as a package for the R language. Two user-interfaces are provided; PAST can be run by loading the package in R and calling its methods, or by using an R Shiny guided user interface. In testing, PAST completed analyses in approximately half an hour to one hour by processing data in parallel and produced the same results as the previously developed method. PAST has many user-specified options for maximum customization. Thus, to promote a powerful new pathway analysis methodology that interprets GWAS data to find biological mechanisms associated with traits of interest, we developed a more accessible, efficient, and user-friendly tool. These attributes make PAST accessible to researchers interested in associating metabolic pathways with GWAS datasets to better understand the genetic architecture and mechanisms affecting phenotypes.

## 1. Introduction

Genome-wide association study (GWAS) of complex traits in maize and other crops has become very popular for identifying regions of the genome that influence these traits [1,2,3]. In general, hundreds of thousands of single nucleotide polymorphisms (SNPs) markers are each tested using F statistics for an association with the trait, which assigns a *p-*value for the SNP-trait association. Individual marker-trait associations that meet the threshold set for the false discovery rate (FDR, the proportion of false positives among all significant results for some level α) are then studied in more detail to uncover hints as to the genetic architecture of the trait, and how best to improve it in the future. Many true associations may be missed in GWAS, because the threshold for FDR could be as low as α divided by the total number of SNPs being tested. In complex, polygenic traits, the effects of genes that exert only small effects on a trait may not meet the FDR threshold, especially if the effect value of the association is influenced by the environment. Additionally, alleles of many genes may be expressed only in specific genetic backgrounds and will only be useful when found in combination with the positive alleles of other genes in the same pathway [3]. These allelic combinations may not exist in the limited number of individuals in the GWAS panel. Thus, the statistical power of GWAS for detecting genes of small effect is limited by the strict levels set for FDR and by insufficient numbers of high-frequency polymorphisms found in most panels.

Metabolic pathway analysis focuses on the combined effects of many genes that are grouped according to their shared biological function [4,5,6]. This is a promising approach that can complement GWAS to give clues about the genetic basis of a trait. Originally developed to study differences in gene expression data in human disease studies [7], pathway analysis and association mapping have been used in medical research to find biological insights missed when focusing on only one or a few genes that have highly significant associations with a trait of interest [5,8,9,10]. Pathway analysis has only just begun to be used as well in plant and animal studies [11,12]. In addition, biologically relevant pathways can be used to guide the interpretation of large data sets produced by other high-throughput approaches like RNA sequencing, proteomics, and metabolomics.

More recently, GWAS-based metabolic pathway analysis has been used as a discovery tool to investigate the genetic basis of complex traits in plants. A pathway-based approach was used to study aflatoxin accumulation [13], corn ear worm resistance [14] and oil biosynthesis [15] in maize. Combining GWAS analysis with metabolic pathway analysis considers all genetic sequences positively associated with the trait of interest, regardless of magnitude, and jointly may highlight which sequences lead to mechanisms for crop improvement and which warrant further study and manipulation, for example, by gene editing. While combined GWAS and pathway analyses were highly successful in uncovering associated pathways, the analyses were slow and cumbersome, as the analysis tools were written in a combination of R, Perl, and Bash, and the output of each analysis was manually input into the next analysis. A single, unified and user-friendly tool to accomplish this pathway analysis was lacking.

The Pathway Association Study Tool (PAST) was developed to facilitate easier and more efficient GWAS-based metabolic pathway analysis. PAST was designed for use with maize but is usable for other species as well. It tracks all SNP marker-trait associations, regardless of significance or magnitude. PAST groups SNPs into linkage blocks based on linkage disequilibrium (LD) data and identifies a tagSNP from each block. PAST then identifies genes within a user-defined distance of the tagSNPs and transfers the attributes of the tagSNP to the gene(s), including the allele effect, R^2^ and *p-*value of the original SNP-trait association found from the GWAS analysis. Finally, PAST uses the gene effect values to calculate an enrichment score (ES) and *p*-value for each pathway. PAST is easy to use as an online tool, standalone R script, or as a downloadable R Shiny application. It uses as input TASSEL [16] files that are generated as output from the General Linear or Mixed Linear Models (GLM and MLM) in table format, or files from any association analysis that has been similarly formatted, as well as genome annotations in GFF format, and a metabolic pathways file. The metabolic pathways file should contain one line per gene, and the columns should describe the pathway ID, pathway name, and gene ID.

## 2. Results

PAST is implemented as an R package and is available through Bioconductor 3.10 [17], Github [18], and through MaizeGDB. PAST was based on a method developed by our research group [6]. The original method was subsequently used in two other maize studies [14,15], but required users to customize Perl and R scripts and run Bash scripts. PAST’s implementation is completely in R and requires a user to install the package without needing to edit the source code. Two graphical user interfaces are available in the form of R Shiny applications. A generic version is available on Github and upcoming on CyVerse, while a maize-specific version is planned for MaizeGDB [19] (explained below).

PAST was tested using data from three previous corn GWAS on kernel color (261,147 SNPs), aflatoxin resistance (261,184 SNPs), and linoleic oil production (558,529 SNPs). All three tests were run on a desktop computer with 32 GB of memory, a 4 GHz Intel Core i7 with four processors, and solid-state storage. All four processors were used when testing PAST. The kernel color test completed in ~34 min; the aflatoxin test completed in ~34 min; and the linoleic oil test took ~50 min. Using the previous method, these analyses took 24 h or more, depending on how attentive the user was when starting the next step in the process. The results of the analyses of all three traits were comparable when generated with PAST or with the previous method.

Two versions of an R Shiny application that use PAST have been developed. These R Shiny applications provide a guided user interface that sets analysis parameters in PAST; they can also upload a saved set of results to explore again. The version available on Github and planned for CyVerse allows a user to run a new analysis by selecting their data, annotations, and pathways depending on the species being studied. The version that is available on MaizeGDB [19] allows a user to upload their data and select specific versions of the maize annotation and pathways databases available on MaizeGDB. A screenshot of the generic R Shiny application is provided in Figure 1.

## 3. Discussion

PAST is run by calling its functions with GWAS data from within an R script or by using an included R Shiny interface. PAST will allow a new interpretation of GWAS results, which should identify associated pathways either when one or a few genes are highly associated with the trait (these would have been identified by the GWAS analysis directly); or when many genes in the pathway are moderately associated with the trait (these would not necessarily have been identified by the GWAS analysis). Such an interpretation will add both additional results, and biological meaning to the association data, as was seen with oil biosynthesis in studies by Li et al. [15,20]. While PAST may be useful in bringing biologically useful insights to a GWAS analysis, it will not be able to find order from a chaotic dataset if environmental variation, experimental error, or population structure were not accounted for with the most appropriate analysis model during the association analysis. For strong data sets, however, it may find pathways where GWAS found few or no significant associations which, taken in isolation, shed no real light on the genetic mechanisms underlying the traits under study. PAST may be able to overcome this limitation and may in addition be able to identify epistatic interactions between genes in the same pathway [21], a notoriously difficult thing to do in a GWAS analyses of limited sample size (i.e., a panel of only hundreds of individuals, rather than thousands).

The use of metabolic pathway analysis to derive functional meaning from GWAS results has been used extensively in human disease studies, and methodologies and tools similar to PAST have been published for use with annotated human pathways. Some methodologies reviewed by Kwak and Pan [22] include GATES-Simes, HYST, and MAGMA. Two tools for human GWAS pathway studies have been published: GSA-SNP2 [10] and Pascal (a Pathway scoring algorithm) [23]. However, most of these tools would need to be extensively modified to work with any set of user-supplied pathways outside human studies. In order to compare PAST to two other tools that could be used with user-supplied pathways and genes, MAGMA and INRICH were tested with kernel color data. The trans-lycopene biosynthesis pathway is the most important pathway in this trait because it creates carotenoids, intensely yellow and orange pigments in maize grain. MAGMA, which has a bias towards human analysis, did not report the trans-lycopene biosynthesis pathway as significant when testing the kernel color data. INRICH [24] did not show the bias towards human analysis that MAGMA did, however, due to extreme difficulties getting the data formatted, INRICH could not be tested at all with the grain color example. PAST detected the trans-lycopene biosynthesis pathway correctly. Similarly, the linoleic acid results obtained using MAGMA were not as accurate as PAST, as compared to the previously published results [15]. In addition, a unique function of PAST is its ability to test the pathways associated with an increase in the phenotypic expression of a trait separately from the pathways associated with a decrease in the trait; this cannot be done with other tools tested.

MAGMA and INRICH lack a GUI and require the use of command-line tools. In comparison, running PAST does not require familiarity with command-line tools. For users with some familiarity with the R language, PAST can be run via an R script. For users unable to run R scripts, PAST is available as an R Shiny application that allows them to select their input files and parameters via a guided user interface, something that the tested alternatives lacked entirely.

An analysis with PAST should be illuminating for any plant species, and while it is expected to work better with outcrossing species due to faster linkage disequilibrium breakdown, it has been run successfully with potato and wheat (data not shown). Because inbreeding and polyploid species have very long LD blocks which may contain multiple, equally linked genes, or homology to more than one genome, the assignment of SNPs to genes may be more complicated. Additional tests will be run to see if these problems negate the use of this tool. PAST will also work with any animal and human datasets. The only requirement for a successful PAST analysis is that annotated pathway/genome databases (or related model organism databases) and GFF annotations must be available.

In conclusion, we presented PAST, a tool designed to use GWAS data to perform metabolic pathway analysis. PAST is faster and more user-friendly than previous methods, requires minimal or no knowledge of programming languages, and is publicly available at Github and Bioconductor, and will soon be available on CyVerse and MaizeGDB.

## 4. Materials and Methods

PAST processes data through four main steps. First, GWAS output data is loaded into PAST. This data comes in the form of statistics that reflect the effects of specific loci (e.g., SNPs) with a trait(s) of interest and LD data between loci. Second, the SNPs are associated with genes based on LD and genomic distance between SNPs and genes. Once SNPs are assigned to genes, the allelic effects and *p-*values of the SNPs are then transferred to the genes. The genes and their effects are used to find significant pathways and calculate a running enrichment score, which is plotted in a rugplot for each pathway in the fourth step. A flowchart in Figure 2 shows the process.

### 4.1. Loading Data

During the process of loading data, the GWAS dataset is filtered to account for any non-biallelic data. Any data with more or fewer than two alleles associated with that marker is discarded. Data without an R^2^ value (coefficient of determination of the SNP/trait association) is removed as well, since later calculations rely on the R^2^ value. The effects data (the magnitude of the effect of each SNP allele on the phenotype or trait) is associated with the statistics data in order to collect all data about a marker into a single data frame.

The LD data is filtered to drop rows where the loci are not the same, and then unneeded columns from the TASSEL output are dropped. Only data about the locus, the positions, the sites, the distance between the sites, and the R^2^ value (coefficient of determination for LD) is retained. The remaining data is split into groups based on the locus.

### 4.2. Assigning Genes

Genes were assigned the attributes of linked SNPs according to the method described in Tang et al. [6]. SNPs are parsed into linked groups by identifying all pairs of SNPs with LD data that exceed a set cutoff r^2^ for linkage. SNP blocks occur when multiple SNPs are linked to one SNP in common. SNPs that are only linked to one other SNP are considered singly linked SNPs, and SNPs not linked to any other SNPs are unlinked. In all cases, PAST follows an algorithm to identify one tagSNP to represent all linked SNPs in order to reduce the dimensionality of the dataset and identify which allele effect, p and R^2^ to transfer to the physically linked gene(s). Unlinked SNPs are by default identified as the tagSNP. For SNPs that are linked to a single other SNP, if both have the same effect sign (positive or negative), PAST identifies the one associated with the largest effect (absolute value) as the tagSNP. If the effects are equal, the second (more downstream) SNP is used. If, however, the effect signs are different, the SNP with the lowest *p-*value is used. If the *p*-values are the same and the signs are different, the SNP is labeled as problematic, since no assignment can be made, and no tagSNP is identified; these are dropped from the analysis. (To date, these have fortunately been found to be very rare).

The tagSNP within blocks of SNPs is identified by first counting the number of positive and negative effects in each linkage block. If the number of positive effects is greater, then the SNP with the largest positive effect is chosen. If the number of negative effects is greater, then the SNP with the largest negative effect is noted. Ties between the number of negative and positive effects are broken by checking the sign of the SNP in common defining the block. The tagSNP is then the one with the largest effect and the same sign, and it is marked to indicate the number of SNPs in the block. Once all blocks have been reduced to a single tagSNP, the tagSNP is used to locate the nearby gene(s).

Once tagSNPs have been identified, the annotation files are checked to look for genes within a physical distance window provided by the user. The effect and the *p-*value of the tagSNP is transferred to the gene. The SNP-gene assignments are grouped by gene name, and if more than one SNP block or unlinked SNP is found to be linked to the same gene, each gene is tagged by counting the number of negative effect and the number of positive effect associations in the blocks linked to the same gene. If there are more negative effects, the most negative effect and *p-*value is assigned to the gene. If there are more positive effects, the positive effect and *p-*value is assigned to the gene. If there are more than one equally positive or equally negative effects, the effect with the lowest *p-*value is chosen and assigned to the gene. If there are an equal number of negative and positive effects, the effect with the greatest absolute value is selected. The number of linked SNPs is set to the total number of SNPs (SNPs within blocks plus blocks within genes) linked to that gene. Once all the blocks of genes have been processed, the effects of each gene are used to find significant pathways.

### 4.3. Finding Significant Metabolic Pathways

Significant pathways are found by using a previously described method [4,6,7]. User-input determines the minimum number of genes that a pathway must contain to be retained for processing (to avoid small sample size bias), the number of times the effects data are randomly sampled with replacement to generate a null distribution of enrichment scores (ES), and the pathways database that is being used.

For each gene effects column (observed and randomly sampled), the effects are sorted and ranked from best to worst; whether this is in increasing or decreasing order depends on the trait under study and whether the researcher is interested in pathways associated with an increase in the trait (i.e., yield) or a decrease in the trait (i.e., disease progression). The ES running sum statistic for each pathway increases for genes in the pathway and decreases for genes that are not. The amount of increase for genes in the pathway corresponds to the effect for that gene and is weighted by the absolute value of the effect. The pathway ES is the largest positive value calculated for the running sum statistic.

Pathway significance is determined by comparing the observed ES with the ES for the null distribution. The mean and standard deviation for the null distribution are used to normalize the observed ES so that z scores can be obtained. PAST uses 1000 permutations of the effects values (but allows the user to set the number of permutations) to generate a null distribution of ES values. *p*-Values are computed from the z scores using the (1-pnorm) function. Since multiple hypothesis testing is still a concern, an FDR-adjusted *p*-value (known as *q*-value) is calculated using the *q*-value package in R [25].

### 4.4. Plotting

Based on user input, the pathways can be filtered for significance (either *p*-value or *q*-value), or the top *n* (or all) pathways can be selected. Rugplots for each pathway in the set of significant pathways are plotted as the last step. The X-axis shows the rank of each gene effect value; the Y-axis shows the value of the ES running sum statistic as each consecutive gene effect value is processed. An x-intercept line indicates the highest point of the ES. Small hatch marks at the top of the image indicate the rank position of the effect of all genes in the pathway; every gene in the annotated gene file is ranked from the highest to lowest value, but only the genes in the pathway being plotted are highlighted with a hatch mark. An example rugplot is provided in Figure 3.

## Figures and Tables

**Figure 1 plants-09-00058-f001:**
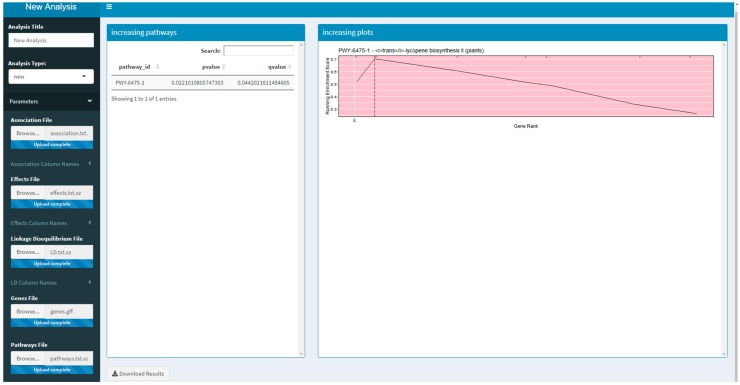
A screenshot of the R Shiny application running the Pathway Association Study Tool (PAST).

**Figure 2 plants-09-00058-f002:**
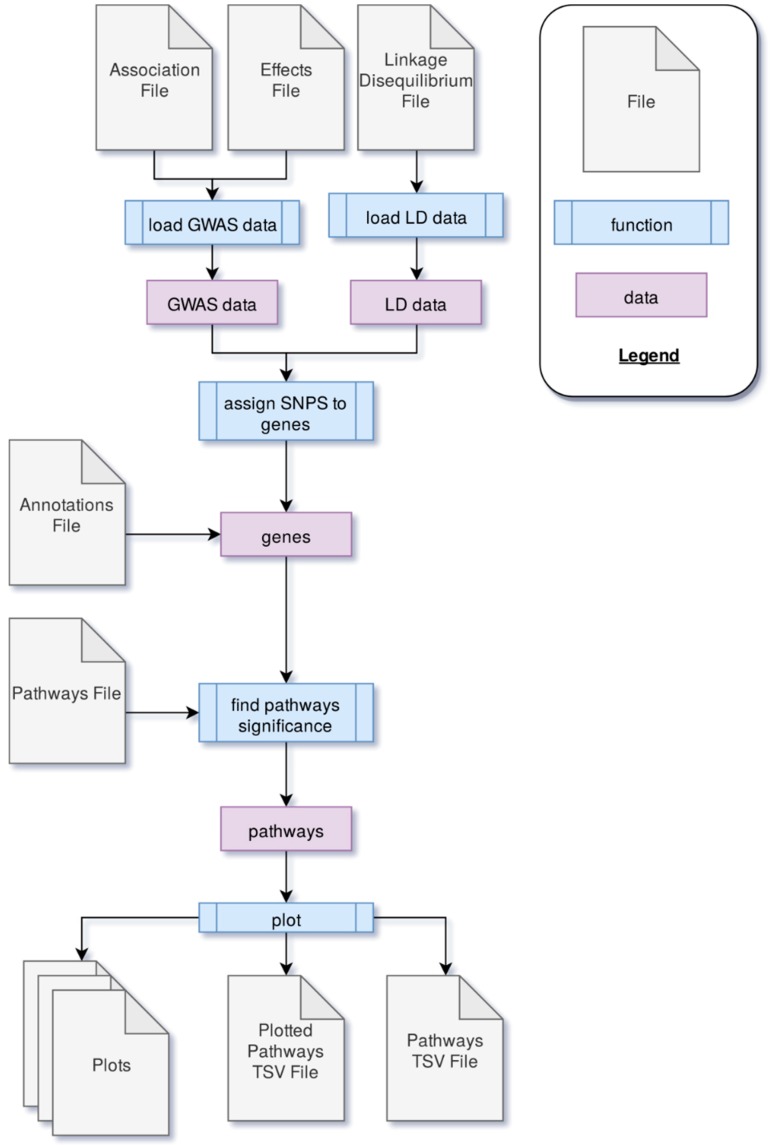
The process through which PAST processes genome-wide association study (GWAS) output data to identify metabolic pathways significantly associated with a trait of interest.

**Figure 3 plants-09-00058-f003:**
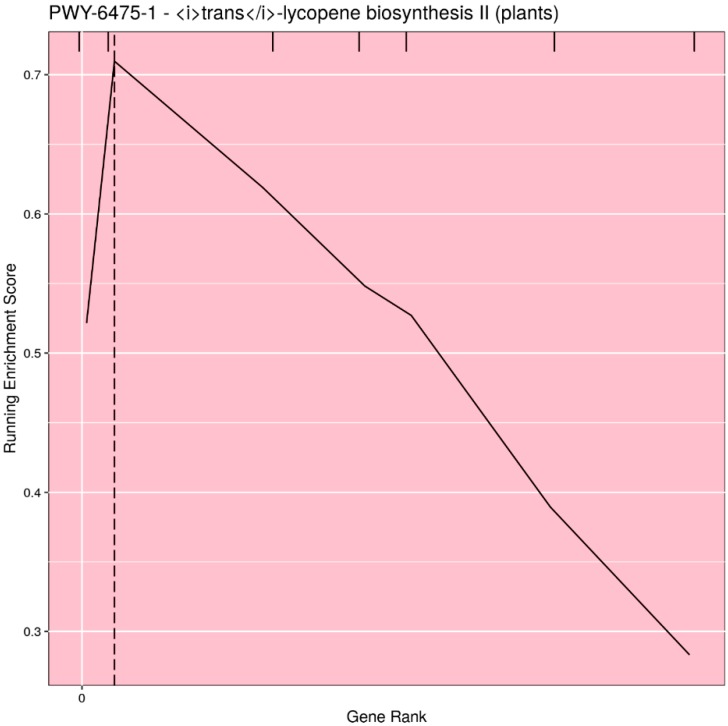
Example of the rugplot graphic generated by PAST for one significantly associated metabolic pathway. The X-axis shows the rank of each gene effect value; the Y-axis shows the value of the enrichment score (ES) running sum statistic as each consecutive gene effect value is processed. The x-intercept line indicates the highest point of the ES. Small hatch marks at the top of the image indicate the rank position of the effect of all genes in the pathway.

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
