# Peer review of "PAST: The Pathway Association Studies Tool to Infer Biological Meaning from GWAS Datasets"

_plants, 2020, doi:10.3390/plants9010058_

Round 1

Reviewer 1 Report

This paper describes the tool PAST, its usage, and applications to GWAS. PAST focuses on pathways rather than single genes, with the goal of expanding the biological interpretations from GWAS results. The
authors correctly note that most similar tools have been developed for human studies, and are not easily applied to plant studies. Tools that are developed for, or easily applied to, plant data sets are very
valuable. The inclusion of a Shiny app is helpful for users who are more comfortable in at GUI environment.

Specific comments:
In the author list, only affiliations 1 and 2 are ascribed to authors. Two authors are lacking affiliations.

Lines 78-81: Which TASSEL file formats are required as input? Figure 2 gives some clue about this, but a little more detail would be very helpful. Are these the _statistics_ and _effects_ files exported in
"table" format? What format is required for the "metabolic pathways" file?

Author Response

1. In the author list, only affiliations 1 and 2 are ascribed to authors. Two authors are lacking affiliations.

This was an error on our part that was introduced when we moved the manuscript to the Plants template and has been corrected.

2. Lines 78-81: Which TASSEL file formats are required as input? Figure 2 gives some clue about this, but a little more detail would be very helpful. Are these the _statistics_ and _effects_ files exported in "table" format? What format is required for the "metabolic pathways" file?

More detail has been added to note that these TASSEL output files are the table output, as well as describing the format of the metabolic pathways file.

Reviewer 2 Report

Is it "Bash" (line 67) or BASH (line 87)?

Line 119: What is an improper analysis model? And what do you consider a proper analysis model? Some guidance may be useful here.

Line 164: Effects and LD between information is forwarded here. What about the precision of effect size estimates? Is this used as well? I think this would be important, much in the spirit of meta-analysis.

Line 176: You mention R^2 here, but it is not obvious how this should be computed with GWAS applications, when a mixed model is used, which is what I would consider a "proper analysis model". There is some recent work on coefficient of determination for (generalized) linear mixed model that the authors could consider here.

Line 182: r2 =>R^2 (check formatting)

Line 233: What is the null distribution for ES? How is this distribution obtained?

Author Response

1. Is it "Bash" (line 67) or BASH (line 87)?

The language should be "Bash". This inconsistency has been corrected.

2. Line 119: What is an improper analysis model? And what do you consider a proper analysis model? Some guidance may be useful here.

More description was added to manuscript to give users an idea of what a proper analysis model might be.

3. Line 164: Effects and LD between information is forwarded here. What about the precision of effect size estimates? Is this used as well? I think this would be important, much in the spirit of meta-analysis.

PAST is an implementation of the method published in Tang et al. 2015. During the development of the method, Dr. Tang did compare weighting versus not weighting the effect value by its precision estimate (F value). Weighting, however, did not give the expected results for the kernel color test gene so she ended up not using that approach. The Tang et al. 2015 paper only reports the pipeline that gave the expected results for the the kernel color test gene.  

4. Line 176: You mention R^2 here, but it is not obvious how this should be computed with GWAS applications, when a mixed model is used, which is what I would consider a "proper analysis model". There is some recent work on coefficient of determination for (generalized) linear mixed model that the authors could consider here.

TASSEL produces a R^2 value using either MLM or GLM outputs. In general, PAST works with input data in tab-separated value (TSV) format and is agnostic as to how the user produces that data, so long as their GWAS results have the appropriate types of data.

5. Line 182: r2 =>R^2 (check formatting)

These issues have been corrected. R^2 is coefficient of determination per marker, and r^2 is coefficient of determination for linkage,

6. Line 233: What is the null distribution for ES? How is this distribution obtained?

A description of how this distribution is obtained has been added to the paper.